# Relaxation of mitochondrial hyperfusion in the diabetic retina via N6-furfuryladenosine confers neuroprotection regardless of glycaemic status

Aidan Anderson[1,9], Nada Alfahad[1], Dulani Wimalachandra[1], Kaouthar Bouzinab[1], Paula Rudzinska[1], Heather Wood[1], Isabel Fazey[1], Heping Xu [2], Timothy J. Lyons [3,4], Nicholas M. Barnes[5], Parth Narendran[6], Janet M. Lord [1], Saaeha Rauz [1,7], Ian G. Ganley [8], Tim M. Curtis[2], Graham R. Wallace[1] & Jose R. Hombrebueno [1,9] ✉

The recovery of mitochondrial quality control (MQC) may bring innovative solutions for neuroprotection, while imposing a significant challenge given the need of holistic approaches to restore mitochondrial dynamics (fusion/fission) and turnover (mitophagy and biogenesis). In diabetic retinopathy, this is compounded by our lack of understanding of human retinal neurodegeneration, but also how MQC processes interact during disease progression. Here, we show that mitochondria hyperfusion is characteristic of retinal neurodegeneration in human and murine diabetes, blunting the homeostatic turnover of mitochondria and causing metabolic and neuro-inflammatory stress. By mimicking this mitochondrial remodelling in vitro, we ascertain that N6-furfuryladenosine enhances mitochondrial turnover and bioenergetics by relaxing hyperfusion in a controlled fashion. Oral administration of N6-furfuryladenosine enhances mitochondrial turnover in the diabetic mouse retina (Ins2^{Akita} males), improving clinical correlates and conferring neuroprotection regardless of glycaemic status. Our findings provide translational insights for neuroprotection in the diabetic retina through the holistic recovery of MQC.

With the incidence of diabetes rising at an alarming rate (615 million people projected worldwide by 2045)[1], the development of therapies aimed at preventing its major complications is of paramount importance. Diabetic retinopathy (DR), a leading cause of blindness in the working-age population, is one of the most prevalent complications, affecting ~80% of people who have had diabetes for 1-2 decades[2]. During the early stages of pathogenesis (non-proliferative DR [NPDR]), microvascular dysfunction and neural retinal degeneration constitute the key pathologies. At the neuronal level, DR is clinically manifested by electrophysiological deficits and visual field (frequency doubling

[1]Institute of Inflammation and Ageing, University of Birmingham, Birmingham, UK. [2]Wellcome-Wolfson Institute for Experimental Medicine, Queen's University Belfast, Belfast, UK. [3]Division of Endocrinology and Diabetes, Medical University of South Carolina, Charleston, SC, USA. [4]Diabetes Free South Carolina, Columbia, SC, USA. [5]Institute of Clinical Sciences, University of Birmingham, Birmingham, UK. [6]Institute of Immunology and Immunotherapy, University of Birmingham, Birmingham, UK. [7]Birmingham & Midland Eye Centre, Birmingham, UK. [8]MRC Protein Phosphorylation and Ubiquitylation Unit, University of Dundee, Dundee, UK. [9]These authors contributed equally: Aidan Anderson, Jose R. Hombrebueno. ✉e-mail: j.m.romero@bham.ac.uk

perimetry) loss, likely due to impairment of photoreceptors, synaptic structures and degeneration of retinal neurons (including amacrine cells and ganglion cells)[3–7]. Such pathology is compounded by damage to Müller cells, the major glial cells in the retina, essential for regulating neurovascular homeostasis[8,9]. NPDR may eventually progress to diabetic macular oedema (DME) and/or proliferative DR (PDR), stages that are underpinned by vascular leakage from damaged blood vessels into the macula, pathogenic retinal neo-vascularization and fibrovascular membrane growth, leading to blindness. Despite the economic and health burden of DR, current treatments are limited, usually target the PDR end-stage and only partially effective[10,11]. Developing a safe and effective therapy for the early management of DR is an urgent unmet clinical need.

Protection of the mitochondrial network, with its pivotal role in cellular homeostasis, has emerged as an attractive therapeutic target in DR[12,13]. Irreversible mitochondrial dysfunction underpins the early pathobiology of DR, which is evidenced in the retina of diabetic people and animal models by several hallmarks. These include an abnormal mitochondrial ultrastructure (e.g., disruption of lamellar cristae and vacuolation), mitochondrial DNA (mtDNA) damage and overproduction of mitochondrial reactive oxygen species (ROS)[12–14]. To limit stress-induced mitochondrial dysfunction, healthy tissues rely on mitochondrial quality control (MQC) mechanisms which regulate the replacement of damaged mitochondria. MQC is achieved by the selective removal of terminally damaged mitochondria via mitophagy, a process regulated by diverse pathways including the well characterised PTEN-induced kinase 1 (PINK1) pathway that is initiated upon dissipation of mitochondrial membrane potential[15]. In turn, mitophagy must be adequately co-ordinated with mitochondrial biogenesis (de novo synthesis of mitochondria). This process, controlled by Pparg coactivator 1 alpha (PGC-1α) and mitochondrial transcription factor A (TFAM), regulates the repair, synthesis and replication of mtDNA, which takes place at the mitochondrial nucleoid[16]. Furthermore, mitochondrial dynamics, which regulate the morphology of the mitochondrial network through fusion and fission, have been shown to be important in facilitating mitophagy by separating damaged mitochondria for autophagic removal[17]. As such, the coordinated interplay between all three processes (mitophagy, biogenesis and mitochondrial dynamics) is deemed essential to safeguard MQC and maintain healthy mitochondrial turnover in tissues.

We recently reported that mitochondrial turnover collapses at severe stages of neurovascular injury in the diabetic retina as a result of impairment of both PINK1-mitophagy and mitochondrial biogenesis, leading to the accumulation of damaged mitochondria[12]. Although mitochondrial dynamics have been shown to be altered in the diabetic retina, reflected by an altered expression of molecular effectors controlling fusion (e.g., Mitofusins) and fission (e.g., Dynamin-related protein 1 [Drp1])[18], the manner in which these processes interplay with mitophagy and biogenesis to control MQC during diabetes remains unknown. Understanding this complex interplay, and whether it can be rescued pharmacologically to protect the diabetic retina, has major clinical relevance in DR prevention and treatment

N6-furfuryladenine (kinetin) and its metabolite N6-furfuryladenosine (kinetin riboside [KR]) have recently been reported as potent PINK1-kinase activators[19,20]. To achieve this action, kinetin is first converted to KR by glycosylation, and later phosphorylated to produce kinetin triphosphate (KTP), which acts as an ATP neosubstrate to amplify PINK1-kinase activity[19]. Although KTP can be metabolized from both kinetin and KR, the latter may have unique therapeutic advantages over its precursor. In this regard, KR was shown capable to amplify PINK1-kinase activity independent of mitochondrial membrane potential loss, whilst kinetin requires the aid of mitochondrial depolarizing agents (e.g., carbonyl cyanide m-chlorophenyl hydrazone [CCCP]) to achieve an equivalent effect[19]. In addition, the glycosylation needed to convert kinetin into KR may comprise a rate-limiting

activation step[19]. Nonetheless, beyond PINK1 target engagement, the effectiveness of KR to provide a real therapeutic benefit through the activation of mitophagy and/or mitochondrial biogenesis is yet to be demonstrated.

Here we show that in the diabetic retina and concomitantly with neuronal deterioration, mitochondria remodel towards hyperfusion, blunting their turnover and causing metabolic stress. By mimicking this process in vitro, we also found that in contrast to its precursor kinetin and to other small-molecules reported to activate PINK1-mitophagy (niclosamide and urolithin-A), KR rescues mitochondrial turnover safely, exerting relevant neuroprotection (regardless of glycaemic status) when orally administered to type-1 diabetic (Ins2^Akita) mice.

## Results

### Mitochondrial hyperfusion underpins retinal neurodegeneration in human diabetes

We reported recently that mitochondrial turnover deteriorates in the murine diabetic retina (Ins2^Akita) concomitantly to retinal neurodegeneration[12]. To understand if such pathology is translatable to human diabetes, we first characterized neurodegenerative hallmarks in the human diabetic retina and its association with abnormal mitochondrial profiles using immunohistochemistry (Fig. 1). Donors included retinas from people with diabetes but no clinical retinopathy and mild NPDR, as well as retinas from non-diabetic donors. We focused primarily on the outer retina (from outer limiting membrane [OLM] to outer plexiform layer [OPL]), since as shown in murine diabetic models (including Ins2^Akita mice) this is the major site for mitochondrial turnover[12,21] and also for key neurodegenerative hallmarks, including photoreceptor and synaptic stress[5,22,23]. Detailed morphometric quantification revealed that mitochondrial networks acquire a hyperfused state in the outer retina of diabetic individuals (Fig. 1a, arrowheads), as evidenced by increased mitochondrial interconnectivity ($P = 0.009$), average length ($P = 0.024$) and aspect ratio ($P = 0.034$) (Fig. 1b) when compared to non-diabetic donors.

Mitochondrial hyperfusion was associated with retinal neurodegeneration in human diabetes (Fig. 1c–j). Quantification of DAPI$^+$ nuclei at the ONL showed no changes in photoreceptor densities (despite showing a bimodal distribution in diabetic individuals Fig. 1c–e). However, antibody against medium-wavelength opsin (M-opsin) revealed atrophy of cone photoreceptor outer/inner segments (Fig. 1d, arrowheads) and of their synaptic terminals (Fig. 1d, arrows). Further analysis using antibody against synaptophysin demonstrated that diabetic individuals present a marked reduction of presynaptic elements at the OPL ($P = 0.046$, Fig. 1f, h, arrowheads). Calbindin immunohistochemistry also demonstrated loss of postsynaptic elements from horizontal cells at the OPL in diabetes (Fig. 1g arrowheads). In the diabetic individuals evaluated (without clinical retinopathy [DNR] and NPDR), the severity of outer retinal deterioration was not apparently associated with the clinical stage of the disease (Supplementary Fig. 1), but rather with the level of mitochondrial fusion, including photoreceptor ($r^2 = 0.4$, $P = 0.049$ Fig. 1i) and presynaptic loss ($r^2 = 0.64$, $P = 0.005$ Fig. 1j). Taken together, our data suggest that altered mitochondrial dynamics (specifically hyperfusion) may contribute to the pathobiology of retinal neurodegeneration in human diabetes.

### Mitochondrial hyperfusion blunts mitochondrial turnover in retinal Müller glia in experimental diabetes

We next investigated whether compromised dynamics impact mitochondrial turnover in the diabetic retina. To this end, we took advantage of diabetic mitophagy-reporter mice (mitoQC Ins2^Akita)[12], since MitoQC allows for the reliable evaluation of mitochondrial morphology along with mitophagy via the targeting of a tandem mCherry-GFP tag to mitochondria using a signal peptide derived from the protein Fis1 (Fig. 2a). Furthermore, Ins2^Akita mice have the added value of closely

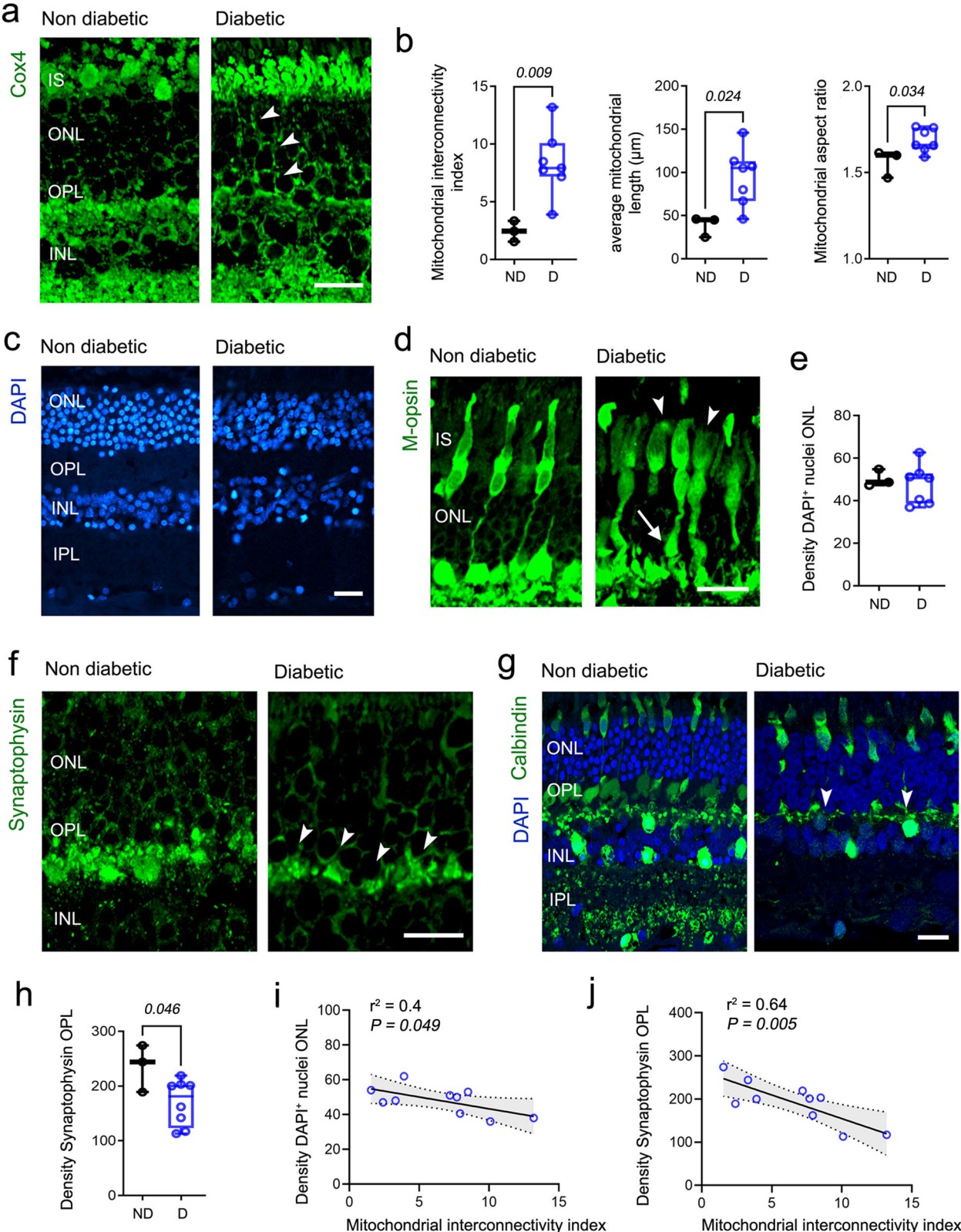

recapitulating human neurodegenerative features found in human diabetes (Fig. 1c–g), as previously reported by our group[5,22,23]. Analysis was performed at stages of mild to severe retinal neurodegeneration (i.e., at 3-months and 8-months of diabetes, respectively)[5]. No major changes in mitochondrial fusion were observed by 3-months of diabetes compared with age-matched non-diabetic controls (Supplementary Fig. 2). In contrast and resembling the human diabetic retina,

mitochondrial fusion was significantly exacerbated by 8-months of diabetes throughout the ONL (Fig. 2b, c, arrowheads). This phenotype was corroborated in non-mitophagy diabetic reporter mice (Ins2^Akita) by similar morphometric analysis using immunostaining against TOMM20 (a translocator of the outer mitochondrial membrane) (Supplementary Fig. 3a–c), and as also evidenced by increased levels of Mitofusin-2 in retinal lysates (Mfn2) (Supplementary Fig. 3d).

**Fig. 1 | Neuronal degeneration is associated with mitochondrial hyperfusion in the human diabetic retina. a, b** Mitochondrial fusion was evaluated at the outer nuclear layer (ONL) of nondiabetic (ND) and diabetic (D) human donors using Cox4 immunostaining. Arrowheads indicate hyperfused mitochondrial networks. **b** Morphometric quantification of mitochondrial fusion (interconnectivity, average mitochondrial length and aspect ratio). Retinal micrographs from nondiabetic and diabetic human donors processed for **c** DAPI and **d** medium-wavelength opsin (M-opsin). Atrophy of cone photoreceptor outer/inner segments (arrowheads) and of synaptic terminals (arrows). **e** The density of DAPI+ nuclei at the ONL. Retinal micrographs from nondiabetic and diabetic human donors immunostained for **f** synaptophysin and **g** Calbindin. Arrowheads indicate loss of **f** presynaptic and **g** postsynaptic elements at the outer plexiform layer (OPL). **h** The density of synaptophysin+ processes at the OPL. Correlation of mitochondrial fusion (interconnectivity index) against the density of **i** DAPI+ nuclei at the ONL or **j** synaptophysin+ processes at the OPL. Regression line, significance levels ($P$), coefficient of determination ($r^2$) and 95% confidence bands of the best fit line (grey) are shown. ND ($n = 3$ donor eyes), D ($n = 7$ donor eyes (**b–e**), $n = 8$ donor eyes (**h**)). (**b–e, h**) Data are presented in box-and-whisker plots with single data points (for definition of boxplot elements see "Methods" section). *P values* were calculated using 2-sided unpaired Student's *t*-test. IS, photoreceptor inner segments; INL, inner nuclear layer; IPL, inner plexiform layer; Scale bars: 40 μm.

Suggestive of a potential interplay between mitochondrial dynamics and turnover, mitophagy was significantly reduced in the Ins2Akita retina by 8-months of diabetes (i.e., reduction of mCherry-only foci in mitoQC Ins2Akita, $P = 0.0016$; Fig. 2d, e). This was also linked with the accumulation of mitochondria at this stage, as evidenced by increased Fis1 contents at the ONL (Fig. 2f, g) and of mitochondrial ATP-synthase in retinal lysates (Fig. 2h). Importantly, mitochondrial hyperfusion strongly correlated with the impairment of mitophagy in diabetes (Fig. 2i, j), suggesting that impaired dynamics may blunt mitochondrial turnover leading to the accumulation of mitochondria in advanced neurodegenerative disease.

To corroborate a causal interplay between mitochondrial dynamics and turnover, we antagonized mitochondrial fission in retinal Müller glia under hyperglycaemia, using a highly specific Drp1-inhibitor peptide, P110[24] (Fig. 3a). Müller glia were selected for this study, since i) they play a major role in mitochondrial turnover, as a major fraction of fused mitochondria and mitolysosomes were found within Müller cell processes at the ONL (glutamine synthase [GS] positive) (Supplementary Fig. 4) and ii) they play a key role in supporting neuronal homeostasis[12,25]. To investigate such interplay with precision, we took advantage of primary Müller cells (PMCs) isolated from mitoQC mice (MitoQC-PMCs, Fig. 3a). Detailed morphometric analyses in these cells grown under hyperglycaemia (HG - 30.5 mM) for 2 consecutive days showed a more fragmented mitochondrial network when compared to normal glucose (NG - 5.5 mM) conditions (Fig. 3b, c). Such mitochondrial fragmentation was linked to increased mitophagy ($P = 0.021$; Fig. 3b, d) and reduced mitochondrial mass ($P = 0.011$; Fig. 3e), suggesting that mitochondrial fission facilitates mitochondrial turnover under hyperglycaemia. This relationship was corroborated through pharmacological inhibition of mitochondrial fission with P110 peptide. Accordingly, and mimicking our in vivo diabetic observations, P110 treatment exacerbated mitochondrial fusion under hyperglycaemia, suppressing mitophagy and increasing mitochondrial mass (Fig. 3b–e). This effect was also corroborated in the human *Moorfields/Institute of Ophthalmology Müller 1* (MIO-M1) cell line, through similar morphometric analysis (Supplementary Fig. 5a, c-e).

Furthermore, recapitulating the complex MQC dysregulation of the diabetic retina, mitochondrial biogenesis effectors evaluated via TFAM+ immunostaining demonstrated deterioration in MitoQC-PMCs (Fig. 3f) and MIO-M1 cultures treated with P110 under hyperglycaemic but not normoglycaemic conditions (Supplementary Fig. 5b, f). Overall, the adaptive mitochondrial fragmentation required to facilitate mitochondrial turnover in diabetes is hampered by mitochondrial hyperfusion in Müller glia during advanced neurodegeneration.

### Mitochondrial hyperfusion causes metabolic stress in Müller glia under hyperglycaemic but not normoglycaemic conditions

Mitochondrial fusion could be an adaptive response to maximize cellular bioenergetics in diabetes[26], so we next investigated how exacerbated fusion impacts mitochondrial function. Indicative of a deteriorated mitochondrial health, we observed a profound dissipation of mitochondrial membrane potential (ψm) in MIO-M1 cells

treated with P110 under hyperglycaemic but not normoglycemic conditions (as evidenced by significantly reduced JC-1 red to green fluorescence ratios – Fig. 3g, h).

These results suggest that decreased mitochondrial turnover, resulting from exacerbated fusion, worsens rather than improves mitochondrial bioenergetics in diabetes. To corroborate this finding, we assessed oxygen consumption rate (OCR) using the *Seahorse* XF96e bio-energetic assay (Fig. 3i–m). Under normoglycaemic conditions, P110 peptide was found to enhance mitochondrial bioenergetics, including basal respiration, maximal respiration and consequently, OCR metabolic potential (i.e., increased capability to meet energy demands via mitochondrial respiration; Fig. 3i–m). However, under hyperglycaemic conditions, P110 fusion peptide significantly deteriorated mitochondrial bioenergetics, as observed by the significant decrease of basal- and maximal-respiration, ATP-linked respiration and OCR metabolic potential (i.e., increased capability to meet energy demands via mitochondrial respiration; Fig. 3i–m). A similar deterioration of mitochondrial bioenergetics was caused by P110 peptide in mouse primary Müller cells under hyperglycaemia (Supplementary Fig. 6), strongly suggesting that exacerbated fusion promotes metabolic stress in Müller glia under diabetic conditions.

### Mitochondrial hyperfusion elicits Müller glia pro-inflammatory stress under hyperglycaemia

We next investigated whether blockade of mitochondrial turnover elicits glial and neuronal DR-like pathology in vitro. A key feature of neuro-inflammatory stress in the diabetic retina involves upregulation of glial intermediate filaments (IFs), notably Vimentin and Glial fibrillary acidic protein (GFAP)[27,28]. In Ins2Akita mice, certain neuroinflammatory components were linked to mitochondrial hyperfusion, as seen by the upregulation of Vimentin throughout the ONL (arrowheads, Fig. 4a, b) (however, no GFAP upregulation towards the outer retina is consistently observed in this model)[22,29]. Confirming the impact of mitochondrial hyperfusion on glial stress, P110 treatment under hyperglycaemic but not normoglycemic conditions elicited Vimentin upregulation in Müller cell cultures (both primary and MIO-M1 cells) (Fig. 4c, d).

Müller glia are an important secretory cell in the retina, where IFs upregulation is usually coupled with the release of pro-inflammatory and pro-angiogenic mediators[8]. As evaluated using multiplex bead array (Luminex), MIO-M1 cultures treated with P110 under hyperglycaemic but not normoglycemic conditions exhibited an elevated secretion of monocyte chemoattractant protein-1 (MCP-1), vascular endothelial growth factor A (VEGF-A) and granulocyte-macrophage colony-stimulating factor (GM-CSF; Fig. 4e). Other pro-inflammatory mediators underpinning DR (e.g., interleukins (IL)-6 and IL-8) were not significantly affected by exacerbated mitochondrial fusion under hyperglycaemia (Fig. 4e).

To further understand the impact of mitochondrial fusion in retinal neuronal stress (particularly at the synaptic level), we took advantage of the potential of Müller stem-cells to differentiate into neuronal lineages[30]. To this end, MIO-M1 Müller cells were induced towards neuronal differentiation using an adapted protocol from

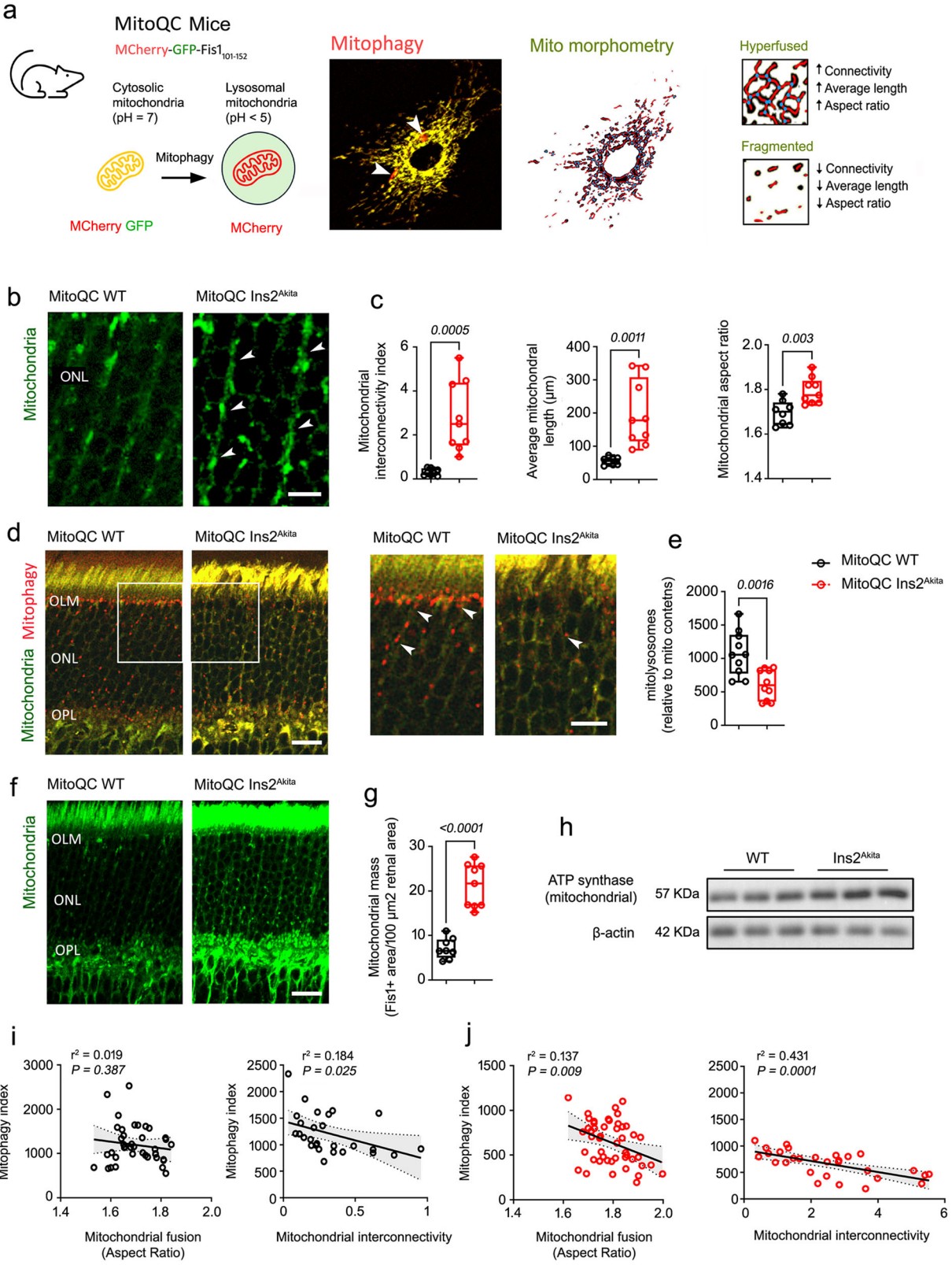

Lawrence et al. (2011)[30]. Neuronal differentiation occurred rapidly (i.e., within 48 h), as confirmed by the expression of β-III tubulin, heavy-chain neurofilament and by the development of complex neurite networks (Fig. 4f). Neurite integrity was not affected by hyperglycaemia, while P110 under normoglycemia caused marginal (yet not significant) retraction (Fig. 4g, h). In contrast and resembling synaptic atrophy in the diabetic retina (Fig. 1d–g), neurites significantly degenerated in P110-treated cultures under hyperglycaemia (arrowheads, Fig. 4g, h). Therefore, increased mitochondrial turnover seems necessary to safeguard neuroglial homeostasis in diabetic conditions; however, this response is compromised by exacerbated mitochondrial fusion.

**Fig. 2 | Mitochondrial hyperfusion underpins impaired mitophagy in the retina of 8-month diabetic Ins2$^{Akita}$ male mice. a** Quantification of mitophagy (mCherry-only-foci, arrowheads) and mitochondrial morphology (Fis1-GFP) using the Mito-QC reporter. **b, c** Mitochondrial fusion was evaluated using Fis1-GFP signal at the outer nuclear layer (ONL) of 8-month diabetic mitophagy reporter mice (mitoQC Ins2$^{Akita}$) and age-matched non-diabetic siblings (mitoQC WT). Arrowheads indicate hyperfused mitochondrial networks. **c** Morphometric quantification of mitochondrial fusion (interconnectivity, average mitochondrial length and aspect ratio) in MitoQC WT ($n = 8$ eyes) and MitoQC Ins2$^{Akita}$ ($n = 9$ eyes). **d, e** Mitolysosome density (arrowheads) at the OLM-OPL of mitoQC WT mice and mitoQC Ins2$^{Akita}$ ($n = 10$ eyes/strain). **f, g** Morphometric quantification of mitochondrial mass (% of Fis1$^+$ signal) at the ONL of MitoQC WT ($n = 8$ eyes) and MitoQC Ins2$^{Akita}$ ($n = 9$ eyes). **h** Example immunoblot of mitochondrial ATP-synthase and β-actin loading controls in retinal lysates of 8-month diabetic Ins2$^{Akita}$ mice and age-matched WT siblings ($n = 3$ eyes/strain). Correlation between mitophagy and mitochondrial fusion (aspect ratio and mitochondrial interconnectivity) in **i** mitoQC WT and **j** mitoQC Ins2$^{Akita}$ mice. Regression line, significance levels (*P*), coefficient of determination (r$^2$) and 95% confidence bands of the best fit line (grey) are shown. **i** MitoQC WT ($n = 25$ [aspect ratio], $n = 27$ [mito interconnectivity]) retinal sections from n = 8 eyes. **j** MitoQC Ins2$^{Akita}$ ($n = 27$ [aspect ratio], $n = 29$ [mito interconnectivity]) retinal sections from $n = 8$ eyes. Data are presented in box-and-whisker plots with single data points (for definition of boxplot elements see "Methods" section). *P-values* were calculated using 2-sided unpaired Student's t-test. OLM outer limiting membrane, OPL outer plexiform layer. Scale bars: 40 μm (**d**, **f**), 20 μm (**b** and inset in **d**).

## Kinetin riboside rescues mitochondrial turnover and bioenergetics in Müller glia by overcoming diabetes-induced hyperfusion

Our Müller cell in vitro platform recapitulated the complex MQC interplay of the diabetic retina (Figs. 3–4), offering a unique opportunity to identify pharmacological compounds capable of rescuing mitochondrial turnover by overcoming hyperfusion in vivo. We previously reported that PINK1-dependent mitophagy deteriorates in Ins2$^{Akita}$ retina by 8-months of diabetes[12]. Therefore, we tested the therapeutic potential of different PINK1-kinase activators to rescue mitochondrial turnover and bioenergetics in the context of diabetes-induced hyperfusion. Four chemical entities were selected based on their reported efficacy in activating PINK1-dependent mitophagy; a) two N6-furfuryladenines (kinetin and its glycosylated metabolite kinetin riboside [KR])[19,20]; b) salicylanilides (niclosamide)[31,32]; c) benzocoumarins (urolithin-A)[33,34].

Under normoglycaemic conditions and reflecting its capability to enhance PINK1-kinase activity[19], KR (but not its precursor kinetin), showed the highest efficacy to activate mitophagy in MitoQC-PMC (Supplementary Fig. 7a, d). This effect, achieved in the nanomolar range, occurred regardless of uncoupling damage as KR marginally increased ψm in human MIO-M1 cultures (Supplementary Fig. 7e). Niclosamide was much less effective in activating mitophagy at equivalent dosages (1 μM; Supplementary Fig. 7b, d). Moreover, and as expected from previous reports[32], niclosamide caused the dissipation of ψm indicating a suboptimal safety profile (Supplementary Fig. 7e). Urolithin-A, despite being reported as a potent mitophagy inducer[33,34], did not activate mitophagy in MitoQC-PMCs even at high concentrations (50 μM; Supplementary Fig. 7c, d).

In the diabetes-mitochondrial hyperfusion context, KR was also capable of activating mitophagy at nanomolar concentrations. This was evidenced in MitoQC-PMCs under hyperglycaemia + P110, where KR (but not vehicle control) relaxed mitochondrial hyperfusion and increased mitophagy in a concentration-dependent manner (Fig. 5a-d). Interestingly, kinetin showed some effectiveness in relaxing mitochondrial hyperfusion, although it had no impact to activate mitophagy (Fig. 5a–d). Moreover, reflecting unique properties in rescuing mitochondrial turnover, KR restored TFAM levels (Supplementary Fig. 8). This effect was not achieved with kinetin or any of the other small molecules tested. Indeed, niclosamide profoundly depressed TFAM expression, suggesting further damage to the mitochondrial biogenesis machinery in the presence of diabetes-induced hyperfusion (Supplementary Fig. 8).

At the functional level, the rescue of mitochondrial turnover by KR was reflected in improved mitochondrial health and bioenergetics. Accordingly, KR but not kinetin restored ψm in human MIO-M1 cultures under hyperglycaemia + P110 (Fig. 5e). Further analysis of metabolic flux using Seahorse, showed that KR rescued bioenergetics in the context of diabetes-mitochondrial hyperfusion, by enhancing basal-respiration ($P < 0.0001$), ATP-linked respiration ($P < 0.0001$) and OCR metabolic potential ($P = 0.025$) (Fig. 5f–j). In contrast, niclosamide and Uro-A aggravated metabolic stress by causing further loss of ψm under hyperglycaemia + P110 (Fig. 5e), confirming their suboptimal safety profile (Supplementary Figs. 7-8). In conclusion, N6-furfuryladenines, particularly in their glycosylated form (e.g., KR), may offer unique therapeutic potential to rescue MQC in the diabetic retina.

## Kinetin riboside rescues mitochondrial turnover in the murine diabetic retina conferring neuroprotection regardless of glycaemic status

Finally, since retinal neurodegeneration was strongly associated with mitochondrial hyperfusion in human diabetes (Fig. 1), we investigated the therapeutic potential of KR in vivo. To circumvent long-term treatment distress in diabetic mice, we opted to administer KR in the drinking water. Effective dosage was established at 60 mg/L (equalling 90 mg/Kg/day), since this dose rescued mitophagy and mitochondrial hyperfusion when delivered to 7.5-month-old diabetic mitoQC Ins2$^{Akita}$ mice for 2 weeks (Fig. 6a–f). TFAM+ mitochondrial nucleoids were concomitantly upregulated, suggesting that KR rescues mitochondrial turnover by also activating biogenesis in the diabetic retina (Fig. 6b, g). The bioactivity of KR was confirmed in Ins2$^{Akita}$ mice when administered long-term (from 4- to 8-months of diabetes), since KR but not vehicle control increased TFAM$^+$ mitochondrial nucleoids in the diabetic retina (arrowheads, Supplementary Fig. 9).

Long-term administration of KR (from 4- to 8-months of diabetes) did not improve other health features of diabetes, namely body weight, blood glucose (invariably above 30 mmol/L) or HbA1c (≥ 110 mmol/mol) levels in Ins2$^{Akita}$ mice (Fig. 6h, i). Nonetheless, concomitant with improved mitochondrial turnover, KR treatment markedly prevented retinal neurodegeneration by 8-months of diabetes. This was evidenced by in vivo spectral-domain optical coherence tomography (SD-OCT) analysis of retinal degeneration, where KR but not vehicle control significantly prevented overall retinal thinning (from photoreceptor segments to nerve fibre layer; arrow Fig. 6j). Assessment of retinal function by electroretinography (ERG), demonstrated that KR preserved photoreceptor function in Ins2$^{Akita}$ mice, as reflected in increased scotopic a-wave amplitudes at higher flash intensities (2.5–25 cd·s/m$^2$; Fig. 6k). Also, suggesting protection of inner retinal function, ERG scotopic b-wave amplitudes were significantly increased at higher flash intensities (8-25 cd·s/m$^2$; Fig. 6l).

The neuroprotective potential of KR was confirmed by a detailed immunohistochemical study (primarily focused on the neuronal pathology described in human diabetes, Fig. 1). At the outer retina, KR but not vehicle control preserved the length of cone photoreceptor segments in Ins2$^{Akita}$ mice (cone-arrestin; Fig. 6m). Synaptic degeneration, one of the main pathological features of human diabetes at the outer retina (Fig. 1d, f, g), was also mitigated by KR. This involved both (i) presynaptic elements from photoreceptors (as shown by a marked

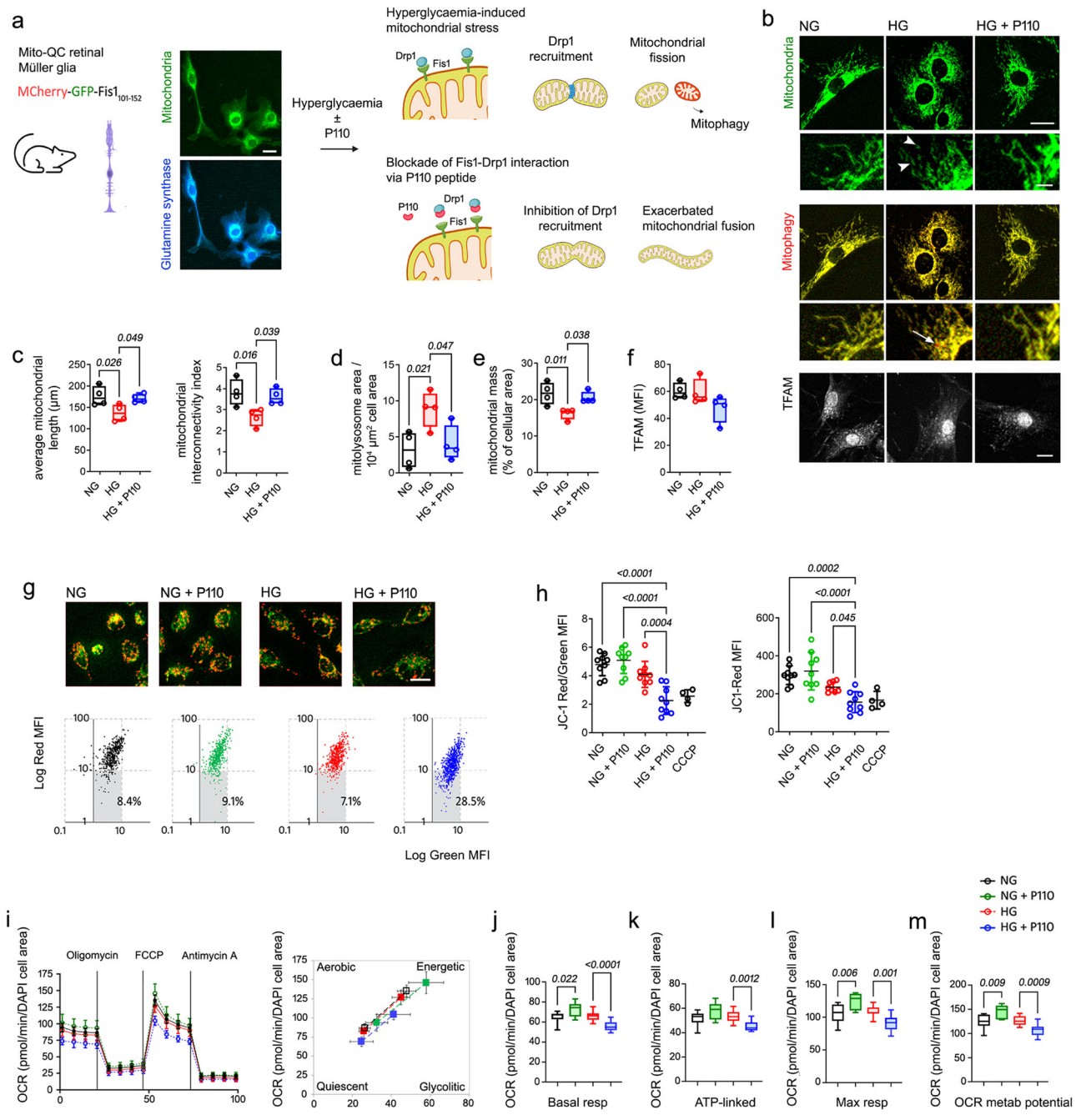

**Fig. 3 | Exacerbated fusion blunts mitophagy impairing mitochondrial fitness in Müller glia under hyperglycaemia. a–f** Primary Müller cells isolated from MitoQC mice (glutamine synthase immunoreactive) or **g–m** human MIO-M1 Müller cells, were antagonized for mitochondrial fission (P110 peptide) under physiological (NG; 5.5 mM) or elevated glucose (HG; 30.5 mM) conditions. Quantification of **b**, **c** mitochondrial fusion (interconnectivity and average mitochondrial length; arrowheads indicate fragmented mitochondria), **b**, **d** mitophagy (arrow), **b**, **e** mitochondrial mass (Fis1-GFP) and **b**, **f** TFAM expression (mean fluorescence intensity [MFI]). **c–f** $n = 4$ independent replicates. **g**, **h** Evaluation of mitochondrial membrane potential (ψm) by JC-1 dye (red, hyperpolarized; green, depolarized mitochondria). CCCP (100 μM) was added as a mitochondrial uncoupler positive

control (2 hours). Gray squares in scatter-plots in **g** indicate % of cells with low ψm. $n = 4$ (CCCP), $n = 9$ (all other groups) independent replicates. **i** Representative Seahorse assay of metabolic flux (left panel) and metabolic potential (right panel) using *Cell Mito Stress Test*. (**j–m**) Quantification of oxygen consumption rate (OCR) indicative of **j** basal respiration, **k** ATP-linked respiration, **l** Maximum respiration and **m** metabolic potential. **j–m** $n = 6$ (NG + P110), $n = 13$ (NG, HG), $n = 14$ (HG + P110) independent replicates. Data are presented as (**c–f**, **j–m**) box-and-whisker plots (for definition of boxplot elements see "Methods" section) or **h**, **i** mean ± SD. *P-values* were calculated using One-way ANOVA with Dunnett's multiple comparison. ECAR extracellular acidification rate. TFAM mitochondrial transcription factor A. Scale bars: 20 μm (**a**, **b**, **g**), 5 μm (insets in **b**).

positive trend in the density of synaptophysin$^+$ terminals at the OPL; Fig. 6n) and (ii) postsynaptic terminals from horizontal cells (calbindin; Fig. 6o). Neuroprotection also extended to the inner retina, as GABAergic amacrine cells were preserved fully by KR in Ins2^Akita mice

(Fig. 6p). Therefore, long-term administration of KR safely restores mitochondrial dynamics and turnover in the diabetic retina, exerting relevant neuroprotection independent of an improved glycaemic status.

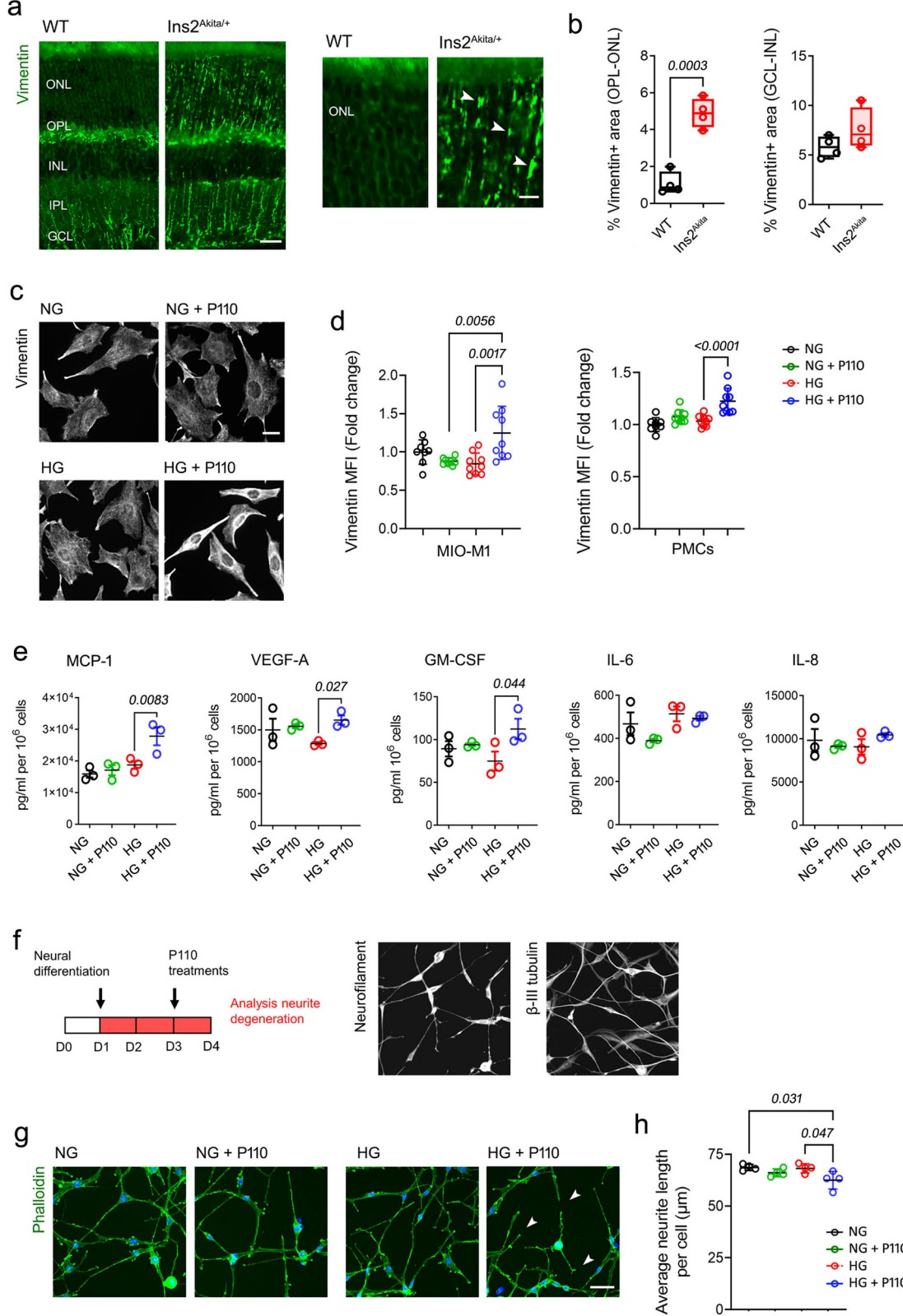

## Discussion

Restoration of mitochondrial quality control (MQC) has emerged as a promising strategy to confer neuroprotection in a large number of conditions, including DR. However, the success of potential interventions depends upon our understanding of how MQC mechanisms (dynamics, mitophagy, and biogenesis) coordinate and are affected during disease progression in specific retinal cells. To date, this knowledge has been constrained by the lack of in vivo models to accurately assess such a complex interplay, but also by our limited understanding of how specific neural components are affected in the human diabetic retina.

The recent development of next-generation reporters including Mito-QC[12,35] has offered a unique opportunity to understand how mitochondrial turnover is impaired in the disease context, and

**Fig. 4 | Exacerbated mitochondrial fusion triggers Müller glia neuroinflammation and neuronal synaptic stress under hyperglycaemia. a, b** Retinal micrographs and quantification of Vimentin radial processes at the outer (ONL-OPL) and inner (GCL-INL) retina of 8-month diabetic mice (Ins2^Akita males) and age-matched WT male siblings. Arrowheads indicate Vimentin processes at the ONL (*n* = 4 eyes/strain). **c, d** Vimentin expression (mean fluorescence intensity [MFI]) in human MIO-M1 cultures (shown in **c**) or mouse primary Müller cells (PMCs), antagonized for mitochondrial fission (P110 peptide) under physiological (NG; 5.5 mM) or elevated glucose (HG; 30.5 mM) conditions. **d** MIO-M1: *n* = 8 (NG, NG + P110), *n* = 9 (HG, HG + P110) independent replicates; PMCs *n* = 8 (NG), *n* = 9 (NG + P110, HG and HG + P110) independent replicates. **e** Luminex® Multiplex Assay of secreted MCP-1, VEGF-A, GM-CSF, IL-6 and IL-8 levels in MIO-M1 cell supernatants grown under NG or HG ± P110 peptide (*n* = 3 independent replicates). **f–h** Human

MIO-M1 cells were induced for neuronal differentiation (48 h) and thereafter cultured for 24 h under NG or HG ± P110 peptide. **f** Validation of neuronal phenotype in MIO-M1 differentiated cultures via expression of heavy-chain neurofilament, β-III tubulin and the development of complex neurite networks. **g, h** Quantification of neurite length in different treatment groups. Arrowheads indicate neurite retraction (*n* = 4 independent replicates). Data are presented as **b**) box-and-whisker plots with single data points (for definition of boxplot elements see "Methods" section) or **d, e, h** mean ± SD. *P-values* were calculated using **b** 2-sided unpaired Student's t-test or **d, e, h** One-way ANOVA with Dunnett's multiple comparisons. ONL outer nuclear layer, OPL outer plexiform layer, INL inner nuclear layer, IPL inner plexiform layer, GCL ganglion cell layer. Scale bars: 60 μm (**a**), 40 μm (**g**), 20 μm (**c** and inset in **a**).

consequently, in the strategic development of new therapies aimed at rescuing MQC safely. Taking advantage of these innovative models (i.e., MitoQC-Ins2^Akita) and challenging the precedent paradigm, we demonstrated that mitochondrial hyperfusion underpins the pathobiology of DR, abrogating mitochondrial turnover at severe neurodegenerative stages. N6-furfuryladenines, particularly in their glycosylated form (i.e., KR), displayed unique capability to restore mitochondrial turnover by overcoming hyperfusion in Müller glia, improving bioenergetics and conferring relevant neuroprotection independent of an improved glycaemic status.

A major strength of our study involves the identification of neurodegenerative changes in the human diabetic retina and their strong association with hyperfused mitochondrial networks. Neurodegenerative hallmarks were found particularly at the outer retina, as evidenced by photoreceptor segment atrophy and profound synaptic loss. Despite the paucity of histological studies, increasing evidence highlights retinal neurodegeneration as a major pathological component in human diabetes. At the clinical level, our findings are supported by electrophysiological studies showing abnormal (cone) photoreceptor function at early-disease stages[3,36]. Psychophysical techniques (colour contrast discrimination tests) have also revealed defects in the cone photoreceptor pathway, as denoted by deficits in the tritan-like axis in all stages of diabetes, including individuals with no signs of retinopathy, indicating abnormalities at the receptoral (short wavelength-cones) and/or post-receptoral (synaptic) levels[37,38]. In support of the latter and adding to our histological evidence, proteomic studies have also demonstrated dysregulated expression of synaptic effectors in the human diabetic retina[39]. Therefore, the protection of photoreceptors and their synaptic machinery appears fundamental in neuroprotective therapies, where protecting mitochondria via restoring their turnover may offer innovative solutions (given the importance of these organelles in supporting phototransduction and synaptic transmission)[40,41]. The use of pre-clinical models recapitulating the neuronal pathology of human diabetes (e.g., Ins2^Akita mice)[5,22], is central to the translational development of such therapies.

Mitochondrial hyperfusion in the outer retina may represent an attempt to optimize mitochondrial function and/or to minimize damage in the context of diabetes. Mitochondrial fusion is known to maximize bioenergetic capacity and/or to counteract mitochondrial stress, through the spreading of healthy mtDNA, metabolites and other helpful components within the meshwork[26,42]. Therefore, therapies that merely promote mitochondrial fission and/or at up-regulating mitophagy at the cost of damaging mitochondria (i.e., uncoupling agents) are unlikely to be advantageous. This was supported in our study, where niclosamide worsened the loss of mitochondrial membrane potential under diabetes-induced hyperfusion. Instead, approaches that restore both mitochondrial dynamics (fusion/fission) and turnover (mitophagy and biogenesis) in a controlled fashion, may provide interesting therapeutic avenues to rescue mitochondrial health and confer retinal neuroprotection in diabetes. N6-

furfuryladenine(s) may meet these criteria, by tuning-up mitochondrial turnover and bioenergetics through overcoming diabetes-induced hyperfusion. In line with our results, recent in vitro studies evidenced the potential of KR to amplify mitophagy in a PINK1-dependent fashion over its precursor kinetin[19,20], collectively indicating that N6-furfuryladenine(s), particularly glycosylated or monophosphate analogues (ProTides)[19], would be required to exert relevant bioactivity. Retinal protection by N6-furfuryladenine(s) represents a potentially convenient approach to DR prevention that could be administered orally or topically, independent of the efforts and risks (e.g., hypoglycaemia) of achieving good glycaemic control.

Müller glia are vital for neuronal homeostasis, recently emerging as key mediators of MQCs through the uptake and removal of photoreceptor mitochondria (so called transmitophagy)[43]. Since the outer retina represents the major site for mitochondrial turnover[12,21], protecting Müller glia in diabetes becomes essential to safeguard MQC and protect neurons from stress-induced mitochondrial damage. Interestingly, Müller glia adopted the characteristic neuroinflammatory phenotype in the context of diabetes induced-mitochondrial hyperfusion, as shown by the upregulation of intermediate filaments (i.e., Vimentin) and the release of inflammatory factors, including MCP-1 and VEGF-A (both elevated in the vitreous of DR individuals)[44–46]. The lack of GFAP upregulation in Ins2^Akita mice, a classic hallmark of human DR[6] limited in this model, also suggests that mitochondrial hyperfusion may not impact on certain neuroinflammatory components.

In addition to these neuro-inflammatory hallmarks, dysfunctional mitochondria may expel components such as ROS, mtDNA and mitochondrial peptides into the cytosol where they are recognised as damage-associated molecular patterns (DAMPs) via inflammasomes[47,48]. Therefore, the blockade of mitochondrial turnover in Müller glia in diabetes may have a detrimental impact on photoreceptor function and neuronal survival, while upregulated mitophagy may prevent mitochondrial damage and neuro-inflammatory stress. Although further work is required to confirm, it is likely that KR exerts a neuroprotective role by rescuing MQC in both glial and photoreceptor cells. A major limitation involves the growth of retinal neurons including photoreceptors in vitro. By mimicking the complex mitochondrial interplay in Müller glia under diabetes, our findings further highlight the value of our in vitro approach for the screening of safe therapies aimed at rescuing mitochondrial turnover pharmacologically in the retina, that would otherwise not be achieved by hyperglycaemia alone.

Despite certain limitations including the need to reproduce results in other representative animal models (including type-2 diabetes), our study provides promising evidence for future translational studies confirming the relevance of N6-furfuryladenine(s) in diabetic eye disease. Although bioactivity was shown through oral application in mice, pharmacokinetic and ADME (absorption, distribution, metabolism, and excretion) studies are required for the development and

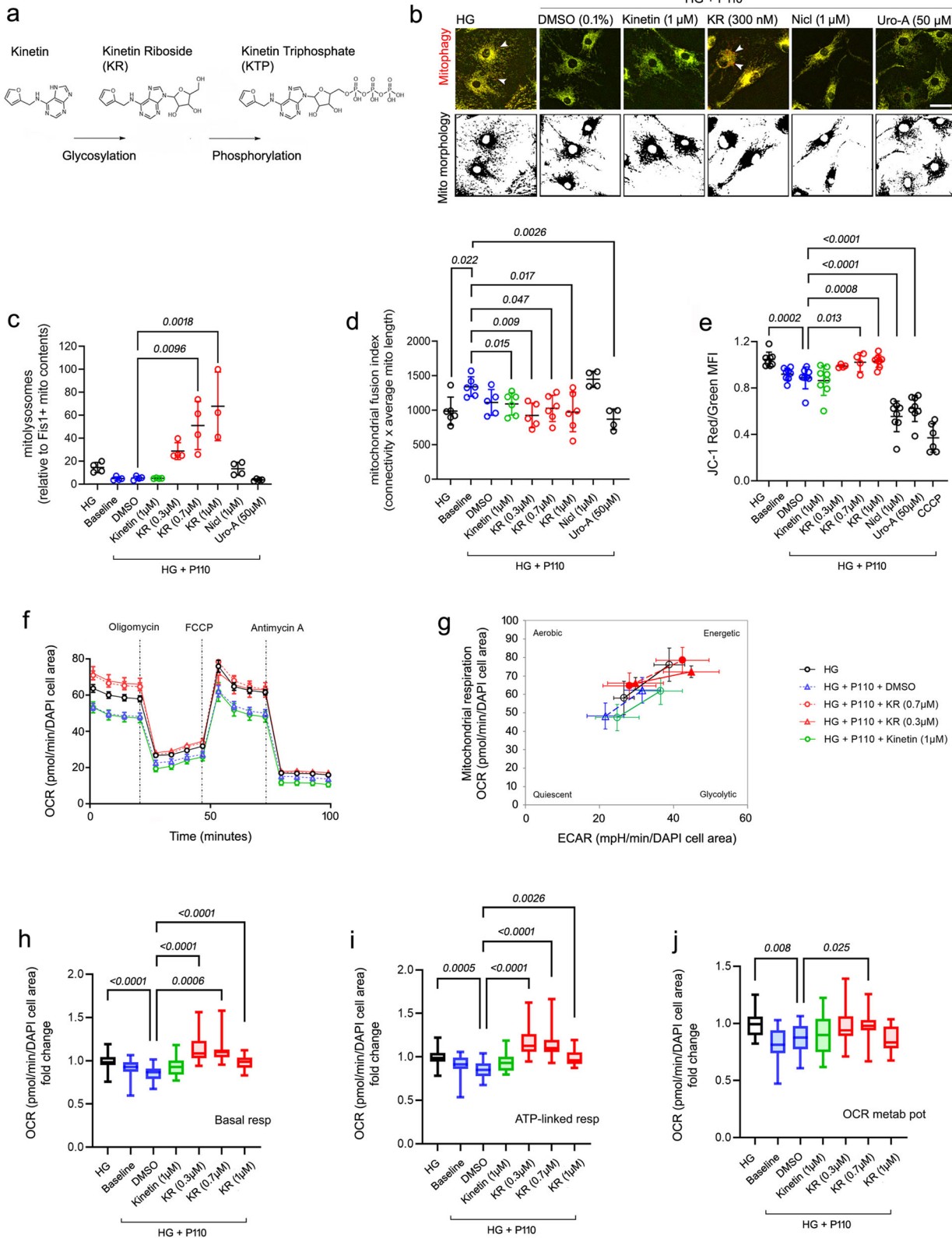

refinement of effective formulations, as well as confirming its safety profile through the absence of systemic side-effects. Combined with recent imaging technology capable of assessing retinal mitochondrial damage non-invasively (Ocumet™)[49], the use of N6-furfuryladenine(s) or advanced analogues may offer innovative opportunities for the early treatment of diabetic eye disease (including eyedrop formulations). The exploitation of our in vitro platform can also bring innovative solutions to discover and repurpose other small-molecules aimed at rescuing mitochondrial exhaustion in diabetes.

## Methods

### Animals

Male heterozygous Ins2$^{Akita}$ (C57BL/6 J) mice (originally obtained from the Jackson Laboratory, Bar Harbor, USA, stock number 003548) and

**Fig. 5 | Glycosylated N6-furfuryladenine (kinetin riboside) rescues mitochondrial turnover and bioenergetics in Müller glia under diabetes-mitochondrial hyperfusion. a** Molecular structure of kinetin and its metabolic conversion into the active substrate kinetin riboside triphosphate. **b**–**d** Primary Müller cells isolated from mitoQC mice or **e**–**j** human MIO-M1 cells were antagonized for mitochondrial fission (P110 peptide) under elevated glucose (HG; 30.5 mM) conditions, and treated with different concentrations of kinetin, kinetin riboside (KR), niclosamide (Nicl), urolithin-A (Uro-A) or DMSO control (0.1%). **b, c** Mitophagy (arrowheads) and **b**–**d** mitochondrial fusion index (as calculated by the product of mitochondrial average length and interconnectivity). **c** $n = 3$ (KR 1 μM), $n = 4$ (all other groups) independent replicates; **d** $n = 4$ (Nicl, Uro-A), $n = 5$ (DMSO, KR 0.3 μM), $n = 6$ (all other groups) independent replicates. **e** Evaluation of mitochondrial membrane potential (ψm) by JC-1 dye (red, hyperpolarized; green, depolarized mitochondria). CCCP (100 μM) was added as a mitochondrial uncoupler positive control (2 hours). $n = 4$ (KR 0.3 μM, KR 0.7 μM), $n = 6$ (CCCP), $n = 8$ (all other groups) independent replicates. Representative Seahorse assay of **f** metabolic flux and **g** metabolic potential using *Cell Mito Stress Test*. Quantification of oxygen consumption rate (OCR) indicative of **h** basal respiration, **i** ATP-linked respiration and **j** metabolic potential. **h**–**j** $n = 13$ (kinetin 1 μM), $n = 14$ (KR 0.3 μM), n = 19 (KR 1 μM), $n = 22$ (HG, Baseline, KR 0.7 μM), $n = 24$ (DMSO) independent replicates. Data are presented as **c**–**g** mean ± SD or **h**–**j** box-and-whisker plots (for definition of boxplot elements see "Methods" section). *P-values* were calculated using One-way ANOVA with Dunnett's multiple comparisons. ECAR, extracellular acidification rate. Scale bars: 40 μm.

age-matched non-diabetic siblings (WT) were used. The Ins2^Akita mice develop severe hyperglycaemia by 4 weeks of age (≥550 mg/dL or 30.5 mM), due pancreatic β-cell loss recapitulating the pathophysiology of type-1 diabetes[50]. MitoQC Ins2^Akita mice were generated by mating mitoQC^+/+ females with Ins2^Akita males[12]. The diabetic phenotype in male offspring was corroborated by blood glucose levels (above 550 mg/dL), while detection of Ins2^Akita and mitoQC-knockin alleles (mCherry-GFP-mtFIS1101–153) by PCR[35]. All mice were kept between 20-24 °C and 45-65% room humidity in standard pathogen-free animal housing rooms with 12/12 h light/dark cycle and access to food (standard chow – EURodent diet 14%, 5LF2; OpCoBe, Sawbridgeworth, UK) and water *ad libitum*.

### Animal treatments
Ins2^Akita mice were treated via supplementation of KR (60 mg/L) or vehicle control (0.1% Dimethylsulfoxide [DMSO]) in the drinking water three times per week (given every other day). Age-matched mice not receiving treatment (Ins2^Akita and WT) served as full controls. Effective KR dosage was inferred via supplementation of KR to MitoQC Ins2^Akita mice from 7.5-months to 8-months of diabetes, by assessing its bioactivity to activate mitochondrial turnover in the diabetic retina (Fig. 6). For long-term treatments, Ins2^Akita mice were treated with KR or vehicle control from 4-months to 8-months of diabetes (Fig. 6). No differences were observed in the amount of water consumed by the diabetic groups (30-35 ml per mouse/day; WT group was 4-5 ml per mouse/day). Mice were randomly assigned to each treatment group.

### Spectral-Domain Optical Coherence Tomography (SD-OCT)
SD-OCT was performed in 8-months diabetic Ins2^Akita mouse groups and age-matched WT mice as previously described[5,51]. Briefly, mice were anesthetized (ketamine 90 mg/kg and xylazine 10 mg/kg) and pupils dilated using 1% tropicamide and 2.5% phenylephrine (Chauvin, Essex, United Kingdom). The Spectralis-Heidelberg OCT system (Heidelberg Engineering, Heidelberg, Germany) was used to evaluate in vivo the total retinal thickness (from nerve fibre layer [NFL] to the photoreceptor inner segments/outer segments [IS/OS]). Retinal thickness was measured at 600 μm eccentricities from the optic disc in dorso-ventral and nasal-temporal regions.

### Electroretinography (ERG)
ERG responses were recorded by 8-months of diabetes in Ins2^Akita mouse groups and age-matched WT mice as previously described[5,22]. In brief, scotopic ERG responses were recorded in anesthetized mice (see above) using a Espion Visual Electrophysiology System (Diagnosys Technologies, Cambridge UK). Scotopic a-wave and b-wave amplitudes were obtained using mouse corneal ERG electrodes (Diagnosys Technologies) in response to single white light-flash intensities (ranging from 0.025–25 cd·s/m²). ERG signals were averaged from 5 responses at each intensity level, with an interstimulus interval of 10 seconds (0.008, 0.025 cd·s/m²), 15 seconds (0.08, 0.25, 0.8 cd·s/m²) or 30 seconds (2.5, 8, 25 cd·s/m²).

### Immunohistochemistry (IHC) of human retinas
Human post-mortem retinas from male and female diabetic donors ($n = 6$ without clinical retinopathy; $n = 2$ with mild NPDR) and non-diabetic donors ($n = 3$, without a history of ocular disease) were obtained from the National Disease Research Interchange (Philadelphia, Pennsylvania, USA). All diabetic donors included in the study did not receive any treatment for DR (e.g., laser photocoagulation and/or intraocular injections). After deparaffinization, retinal sections were washed in PBS and antigen retrieval conducted by immersion (1 hour) in antigen retrieval buffer (EDTA, pH 8.0; Thermo Fisher Scientific) at 60 °C. Sections were then rinsed in PBS and incubated overnight (4 °C) with primary antibodies (Cox4, M-opsin, synaptophysin or Calbindin [Supplemental Table 1]) diluted in 0.5% TritonX-100 with 10% normal donkey serum (NDS) in PBS. Following incubation, retinal sections were rinsed in PBS and incubated for 1 h with appropriate fluorophore-conjugated secondary antibodies (Supplemental Table 2). Retinal sections were cover-slipped with Vectashield/DAPI (Vector Labs, Peterborough, UK) and examined by confocal microscopy (C1 Nikon Eclipse TE200-U, Nikon UK Ltd, Surrey, UK).

### IHC of mouse retinas
Mice were sacrificed by $CO_2$ inhalation and eyes dissected and fixed in 4% paraformaldehyde (Sigma-Aldrich, Dorset, UK) for 2 h. Eyes were then processed for IHC as previously described[52,53]. Briefly, 14 μm-thick retinal cryosections were blocked with 10% NDS in PBS and then incubated overnight (4 °C) with primary antibodies (Supplemental Table 1) diluted in 0.5% TritonX-100 with 10% NDS in PBS. Following incubation, retinal sections were rinsed in PBS, incubated for 1 h with appropriate fluorophore-conjugated secondary antibodies and examined by confocal microscopy as described for human retinas.

### Western blotting
Mice were sacrificed by $CO_2$ inhalation and retinas dissected and lysed in RIPA Lysis Buffer with protease and phosphatase inhibitors cocktails (Sigma-Aldrich). Protein samples (20 μg) were run on 10% (w/v) SDS-PAGE gel and then transferred to PVDF membranes (Millipore, Watford, UK). Samples were then probed for primary antibodies (including β-actin loading controls; Supplemental Table 1) followed by incubation with appropriate HRP-conjugated secondary antibodies, and imaged with G:BOX chemiluminescence system using Syngene's GeneSys software (Syngene; Cambridge, UK). Uncropped western blots scans were provided in the Source Data files.

### Cell culture
The human Müller cell line MIO-M1 was obtained from the UCL Institute of Ophthalmology (London, UK)[54]. PMCs from 3-week old mitoQC^+/+ mice were isolated as previously described[12]. In brief, eyes from euthanized animals were dissected, rinsed with Hanks Balanced Salt Solution (HBSS) and transferred into dispase (2%) in a 5% CO2 incubator at 37 °C for 1 h. Dispase activity was neutralized by washing

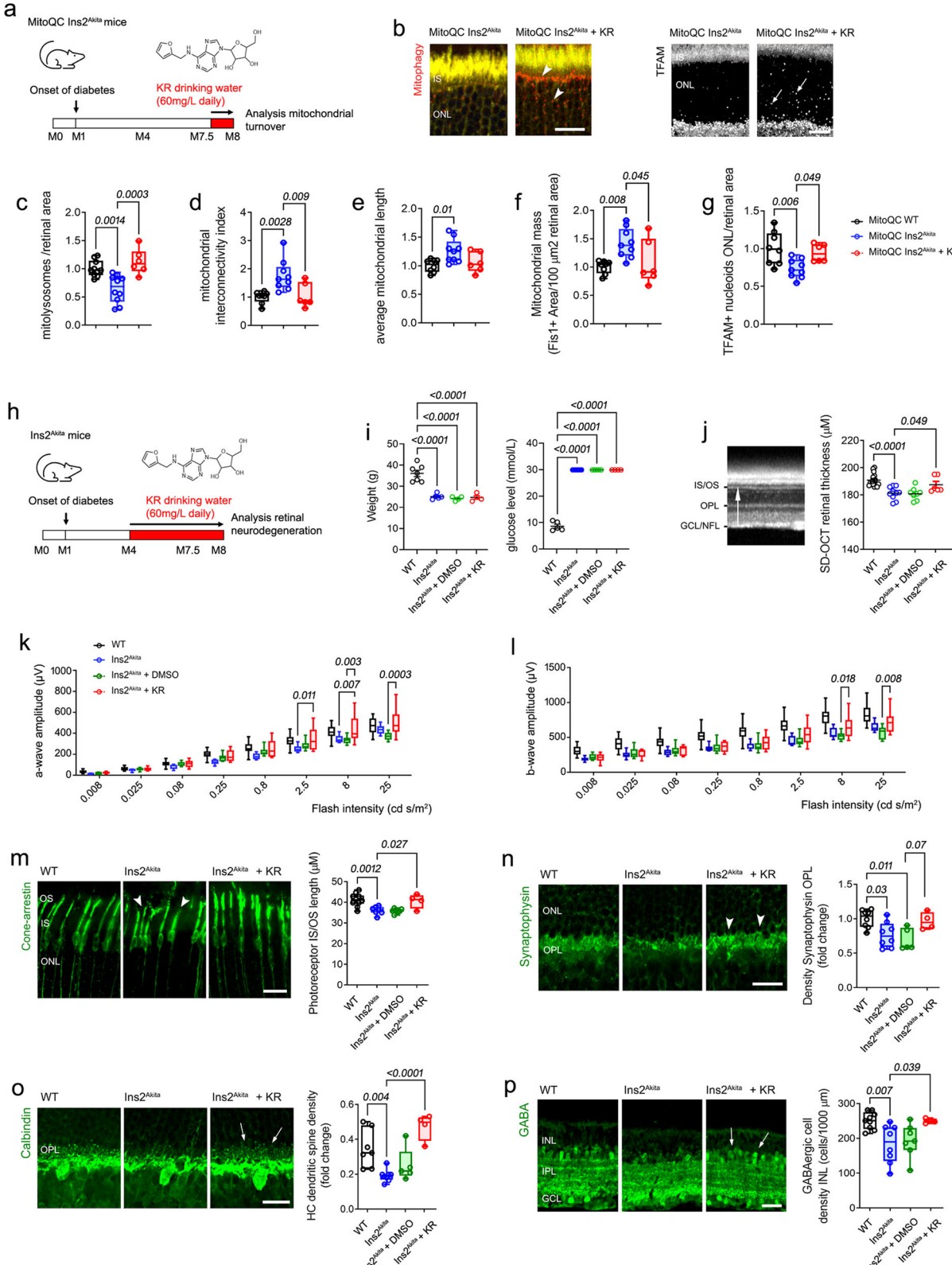

the eyes in Dulbecco's Modified Eagle Medium ([DMEM] 5.5 mM D-glucose; Thermo-Fisher) containing 10% Fetal Bovine Serum [FBS]. After washing, the retinas were carefully dissected, transferred into DMEM 10% FBS, cells dissociated with a pipette and transferred into 6-well plates at 37 °C to grow until confluence. Cells were then trypsinized, expanded in 75 cm2 flask and used for experiments at P2–P4 to avoid variability due to cellular senescence. MIO-M1 and PMCs cultures

were maintained in DMEM containing 10% FBS and 1% Penicillin/Streptomycin (Thermo-Fisher), and supplemented with 5.5 mM D-glucose (NG) or 30.5 mM D-glucose (Sigma-Aldrich; HG) to mimic the hyperglycaemic context in diabetes[12]. Cells were routinely passaged at 1:4 every 3-4 days endpoint experiments performed in 70%–80% confluent cultures. MIO-M1 Cell line and PMCs was authenticated via expression of common Müller glia markers (Glutamine

**Fig. 6 | Kinetin riboside (KR) rescues mitochondrial turnover in the diabetic retina conferring relevant neuroprotection independently of glycaemic status.** a–g Validation of effective KR dosage in vivo. KR was supplemented in drinking water (60 mg/L) to 7.5-month diabetic mitophagy reporter mice (mitoQC-Ins2$^{Akita}$ males) for 2 weeks, and its bioactivity in the diabetic retina confirmed via evaluation of **b**, **c** mitophagy (arrowheads), **d**, **e** mitochondrial fusion (interconnectivity and average mitochondrial length), **f** mitochondrial mass (% of Fis1$^+$ signal) and **b**, **g** TFAM$^+$ mitochondrial nucleoids (arrows). c–g Eyes per strain and condition: mitoQC WT (*n* = 10 [**c**], *n* = 8 [**d-g**]), mitoQC Ins2$^{Akita}$ (*n* = 10 [**c**], *n* = 9 [**d-g**]), mitoQC Ins2$^{Akita}$ + KR (*n* = 6 [c-g]). **h–p** Effective KR dosage (60 mg/L) or DMSO vehicle-control (0.1%) was supplemented in the drinking water from 4-months to 8-months of diabetes in Ins2$^{Akita}$ male mice. Following treatment, glycaemic status and retinal neurodegeneration was evaluated by in vivo and post-mortem approaches. **i** Weight (**g**) and blood glucose levels (mmol/L). Mice per strain and condition: WT (*n* = 7 weight, *n* = 5 glucose levels), Ins2$^{Akita}$ (*n* = 5 weight, *n* = 8 glucose levels), Ins2$^{Akita}$ + DMSO (*n* = 4 weight, n = 5 glucose levels), Ins2$^{Akita}$ + KR (*n* = 4 weight, *n* = 4 glucose levels). (**j**) In vivo quantification of neuroretinal thickness by SD-OCT (from GCL/NFL to IS/OS, arrow). **k**, **l** Retinal function assessed via scotopic electroretinogram (a-wave and b-wave amplitudes). **m** The length of cone photoreceptor segments (cone-arrestin, arrowheads). **n** The density of synaptophysin$^+$ processes at the OPL (arrowheads). **o** The density of horizontal cell dendritic boutons at the OPL (calbindin, arrows). **p** The density of GABAergic amacrine cells at the INL (arrows). j–p Eyes per strain and condition: WT (*n* = 16 [**j**], *n* = 14 [**k, l**], *n* = 10 [**m**], *n* = 8 [**n**], *n* = 7 [**o**], *n* = 9 [**p**]), Ins2$^{Akita}$ (*n* = 10 [**j**], *n* = 6 [**k, l**], *n* = 7 [**m**], *n* = 8 [**n–p**]), Ins2$^{Akita}$ + DMSO (*n* = 8 [**j**], *n* = 6 [**k, l**], *n* = 7 [**m, p**], *n* = 5 [**n, o**]), Ins2$^{Akita}$ + KR (*n* = 6 [**j**], *n* = 8 [**k, l**], *n* = 4 [**m–p**]). Data are presented as (**c–g**, **k–p**) box-and-whisker plots (for definition of boxplot elements see "Methods" section), or (**i**, **j**) mean ± SE. *P-values* were calculated using One-way ANOVA with Dunnett's multiple comparison. IS, photoreceptor inner segments, OS photoreceptor outer segments, ONL outer nuclear layer, OPL outer plexiform layer, INL inner nuclear layer, IPL inner plexiform layer, GCL ganglion cell layer, NFL nerve fibre layer. Scale bars: 40 μm.

synthase, GFAP, Vimentin)[54] and neural stem-cell characteristics (Fig. 4g, h). No mycoplasma was detected in the cell cultures.

## Cell treatments

MIO-M1 or Mito-QC PMCs were seeded onto 96 or 24-well plates and grown overnight before treatment with compounds of interest. To antagonize mitochondrial fission, cells were pre-incubated with P110 Drp1-inhibitor peptide (1 μM; Bio-Techne, Abingdon, UK)[24] in NG for 2 h. Cultures were then washed in PBS and supplemented with P110 (1 μM) in either NG or HG for 72 h. Additionally, other compounds of interest, including kinetin (0.3 - 5 μM, Sigma-Aldrich), KR (0.3–5 μM; Sigma-Aldrich), niclosamide (1 μM; Bio-Techne), urolithin-A (50 μM; Bio-Techne) or vehicle-control (DMSO 0.01%; Sigma-Aldrich) were treated alongside in conditions of NG or HG ± P110. Treatments were changed daily and lasted between 24–72 h depending on the final experimental readout (see below).

## Extracellular metabolic flux analysis

Oxygen Consumption rate (OCR) and extracellular acidification rates (ECAR) were obtained using the Seahorse XFe-96 Flux Analyzer (*Cell Mito Stress Test*; Agilent Technologies, Stockport, UK). MIO-M1 cells and PMCs were seeded at 1200 or 4000 cells/well respectively onto a 96-well microplate, and treated for 72 hours prior to the assay (see *Cell treatments* above). Cells were then washed and medium replaced with Seahorse XF DMEM assay medium (without phenol red) supplemented with glucose (7 mM), sodium pyruvate (1 mM), HEPES (5 mM) and L-glutamine (2 mM). The run consisted of 3 minutes mixing, 3 minutes measuring and subsequent 4 injections as follows: Oligomycin (2 μM), FCCP (0.75 μM), Rotenone (0.5 μM) and Antimycin-A (0.5 μM). Following last Antimycin-A run, Hoechst (Thermo Fisher) was injected (final concentration 1ug/mL) and cell nuclei signal imaged (4X magnification) by epifluorescence microscopy (Olympus IX7 coupled to Retiga R6 CCD digital camera). OCR and ECAR were acquired/analysed using the Agilent Seahorse Wave Desktop software, and flux rate data was normalized to total nuclear area in each well.

## Luminex® Multiplex Assay

Following treatments for 72 h (see *Cell treatments* above), the production of 5 cytokines (MCP-1, VEGF-A, IL-6, IL-8 and GM-CSF) in MIO-M1 cell supernatants were evaluated using the Luminex bead-based assay (Bio-Techne) according to manufacturer's instruction. In brief, 50 μl of the undiluted sample was added to a mixture of colour-coded beads pre-coated with 5 capture antibodies. Following incubation (room temperature for 2 h), 50ul of biotinylated-detection antibodies specific to the analytes of interest were added to the mixture and incubated (room temperature for 1 h). Phycoerythrin (PE)-conjugated streptavidin was then added to the mixture, incubated at room temperature for 30 min and beads read using the Luminex MAGPIX CCD

Imager System (Luminex, Austin, TX). Data was acquired using xPO-NENT software and the levels of each analyte in each sample were normalised against DAPI$^+$ cell counts.

## JC-1 dye staining

The ψm was assessed in human MIO-M1 cells by ratiometric analysis of JC-1 dye (Thermo Fisher). Following treatments for 72 h (see *Cell treatments* above), cultures were supplemented with 0.5 μg/mL JC-1 (30 minutes at 37 °C) and returned to DMEM for live cell epifluorescence microscopy at 37 °C, (Olympus IX7 coupled to Retiga R6 CCD digital camera [QImaging, Teledyne]). Positive controls were supplemented for 2 hours with carbonyl cyanide m-chlorophenylhydrazone (CCCP, 100 μM; Abcam, Cambridge, UK) to uncouple mitochondria. The ψm was assessed by calculating the ratio of mean fluorescent intensity (MFI) between red J-aggregates to green monomers.

## Immunocytochemistry

Cells were fixed in 4% paraformaldehyde (PFA), rinsed in PBS, permeabilised with Triton X-100 (0.5%) and blocked with 10% FBS in PBS. Cells were then incubated overnight (4 °C) with primary antibodies (Supplemental Table 1) diluted in 10% FBS and 0.05% Tween-20 (Sigma-Aldrich) PBS. After incubation, cells were rinsed in PBS, probed with appropriate fluorophore-conjugated secondary antibodies (room temperature for 1 h), washed in PBS and stained with DAPI nuclear dye (1:1000; Sigma-Aldrich) for further microscopy imaging.

## MIO-M1 neuronal differentiation

MIO-M1 Müller cells were induced for neuronal differentiation using an adapted protocol from Lawrence et al (2011)[30]. In brief, cells were plated for 24 h and then supplemented with *PromoCell Neurogenic Differentiation Medium* (C-28015, PromoCell GmbH, Heidelberg Germany). Neuronal differentiation occurred rapidly (48 h), as exhibited by the expression of mature neuronal markers (heavy-chain neurofilament, β-III tubulin) and by the development of complex neurite networks (Fig. 4g). Following differentiation, cells were treated for further 24 h in *PromoCell Neurogenic Differentiation Medium* supplemented with either NG or HG ± P110. Following treatment, cells were fixed in 4% PFA and stained with Phalloidin-FITC (1:1000; Abcam) for further microscopy and morphometric analysis (see below).

## Imaging morphometry

Epifluorescence images were obtained using an Olympus IX7 microscope and Image-Pro Plus software under same acquisition settings. Confocal images were acquired using a C1-Nikon_Eclipse TE200-U microscope and Nikon EZ-C1 software under constant photomultiplier settings. All images were analyzed using FIJI software (v1.54d, National Institutes of Health, Bethesda, MD). To avoid experimental bias during

imaging, retinal regions were selected based on DAPI nuclear signal from middle-center eccentricities. For cell cultures, epifluorescence images (Olympus IX7 coupled to Retiga R6 CCD digital camera) were selected from equivalent cardinal points using bright-field or DAPI nuclear signal. All images were background subtracted prior to analysis.

**Mitochondrial fusion.** The AngioTool plugin[55] (v0.5a) was adapted to evaluate the morphology of the mitochondrial network from Fis1, Cox4 or TOMM20 fluorescence signals. Images from retinal sections (ONL) or individual cells were threshold binarized (constant values were applied for all groups). Images were then assessed for i) average mitochondrial length (equivalent to *average vessel length*); ii) inter-connectivity index (equivalent to *junctions density*); iii) mitochondrial mass (equivalent to *vessels percentage area*). Aspect ratio, an established readout of mitochondrial fusion[56], was also obtained via particle analysis from same images.

**Vimentin and TFAM in human and mouse retinas.** For each particular marker different retinal sectors were threshold and binarized (under constant values), including Synaptophysin (OPL), Vimentin (IS-OPL and inner plexiform layer-ganglion cell layer [IPL-GCL]) and TFAM (ONL). Images were then evaluated by particle analysis and the area of positive signal normalized against the total area analysed.

**Retinal neurodegeneration in human and mouse retinas.** Were quantified in accordance with our previously published methods[5,22,23] including i) density of DAPI⁺ nuclei (ONL), ii) Area of synaptophysin-immunoreactive puncta (OPL), iii) Cone segment length (outer segment plus inner segment), iv) horizontal cell dendritic spine density (OPL) and v) GABAergic amacrine cell density (inner nuclear layer [INL]). Values were normalized to 100 μm retinal length or to the total area analysed.

**Quantification of mitolysosomes (mCherry-only foci) at the outer retina and mitoQC PMCs.** The total mitolysosome area was quantified in accordance with our previously published methods[12]. In brief, GFP was subtracted from the mCherry signal in regions of interest (retina) or individual cells using the "image calculator" plugin of FIJI. The resulting image (mitolysosomes) was threshold binarized (constant values were applied for all groups), and the total mitolysosome area obtained by particle analysis. Values were normalized to mitochondrial mass as obtained from Fis1-GFP signal.

**Quantification of Vimentin and TFAM expression in Müller cell cultures.** Were obtained by the mean fluorescence intensity (MFI) per cell after background subtraction.

**Neurite retraction in differentiated MIO-M1 cultures.** Was obtained by measuring the length of the longest neurite (from soma) in each cell.

### Statistics

For each age group, the difference between 2 means was analysed using 2-sided unpaired Student's *t* test. For comparisons with more than 2 groups, One-way ANOVA (followed by Dunnet's post hoc analysis) was used. ERG responses (scotopic a-wave, b-wave) were analyzed using 2-way ANOVA (followed by Dunnet's post hoc analysis). The animal sample size was adjusted to $n = 4$–10 eyes in each mouse group based on 80% power and a 5% significant level. For retinal analyses, 6-8 images (obtained from at least 2 retinal sections/eye) were evaluated and data averaged per eye for statistical analysis. For Müller cell cultures, at least 40 cells (mitochondrial morphology, mitolysosomes, and neurite retraction) or 200 cells (ψm, TFAM and Vimentin expression) per independent replicate were measured and averaged. N

numbers for and statistical analysis for each readout were specified in Figure legends. Data were expressed as mean ± SEM, as mean ± SD or as box-and-whisker plots, where the line within the box shows the median, box boundaries represent the interquartile range (IQR), and whiskers the minimum and maximum values no more than 1.5 times the IQR. $P < 0.05$ was considered statistically significant. All statistical analyses were performed using Graph Prism software (v10.1.0 - La Jolla, CA).

### Reporting summary

Further information on research design is available in the Nature Portfolio Reporting Summary linked to this article.

## Data availability

All information needed to recapitulate the results presented here can be found in the manuscript, supplementary material and source data. Source data are provided with this paper.

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

## Acknowledgements

This work was supported by *Diabetes UK* (20/0006296) to JRH, *EFSD/Boehringer Ingelheim Grant* to JRH, *Fight for Sight* (1842/1843) to JRH, *Wellcome Trust* (213458/Z/18/Z) to JRH and *Medical Research Council, UK* (MC_UU_00018/2) to IGG. Special thanks to Professor Astrid Limb (University College London) for providing the MIO-M1 cell line. Cartoon in Fig. 3a was created with BioRender.com. We thank the eye donors for their inestimable contribution to diabetes research.

## Author contributions

JRH conceived and designed the project with input from TMC, GW, SR, NMB, HX, PN, IGG and JML. JRH and AA designed the experiments. JRH, AA, NA, DW, KB, PR, HW and IF performed or supervised experiments. JRH, AA, and IF analysed the data. IGG contributed and assisted with Mito-QC experiments. TJL contributed with human retinal samples. JRH wrote and edited the manuscript, with input from all other authors. All authors read and approved the final manuscript for submission.

## Competing interests

The authors declare no competing interests.

## Ethics approval

Human studies were approved by the Oklahoma Health Sciences Centre (OUHSC) and conducted according to the Declaration of Helsinki principles. Written informed consent was received from participants prior to inclusion in the study. Animal studies were approved by the Ethics Committee of the University of Birmingham (Project Licence Number PP6860623) and Queen's University Belfast (Project Licence Number PPL2814). All animal procedures were approved by the Ethical Review Body (AWERB) and authorized under the UK Animals (Scientific Procedures) Act 1986. Animal use conformed to the standards in the Association for Research in Vision and Ophthalmology (ARVO) Statement for the Use of Animals in Ophthalmic and Vision Research and with European Directive 210/63/EU.
