## [Peer Review File · Nature Communications]

Relaxation of mitochondrial hyperfusion in the diabetic retina
via N6-furfuryladenosine confers neuroprotection regardless
of glycaemic statusREVIEWER COMMENTS

Reviewer #1 (Remarks to the Author):

In this paper new mechanisms underlying the important role of mitochondrial remodeling in the pathogenesis of diabetes-induced retinal neurodegeneration are reported. In addition, the neuroprotective effect of the glycosylated form of N6-furfuryladenines (KR) administered orally in Ins2-AKITA mice is clearly shown. This effect was due to the prevention of diabetes-induced mitochondrial hyperfusion of Müller cells regardless the high blood glucose levels. The paper is clearly written, the methodology is correct and the conclusion are based on the results. I do not have mayor concerns on this paper. However, there are minor points that should be addressed:

ABSTRACT

- Line 35 and 37: The word “restored” should be replaced by “prevented”.
- Line 38: “human-relevant” should be replaced by “relevant”. This should be corrected throughout the manuscript.

INTRODUCTION

- The authors could mention the reference (PMID: 17704349) in which the neurodegenerative process was clearly demonstrated in the human retina, and reference (PMID: 30030554) in which a recapitulation of the main events of neurodegeneration were given.

RESULTS

- GFAP overexpression is a characteristic of glial activation in human diabetic retina. I presume that the authors do not measure GFAP in the Ins2-AKITA mice because GFAP overexpression is not found in this model. This is a limiting factor that should be commented in this section, as well as in the discussion.
- Parameters of neural apoptosis should be added to the results
- A direct measurement of KR in the retina Ins2-AKITA mice would be of relevance to strength the results.

DISCUSSION

- The limitations of the study, in particularly the reproducibility of the results in other experimental models and the lack of evaluation of potential systemic toxicity of systemic administration of KR should be commented on in this section.

Reviewer #2 (Remarks to the Author):

Manuscript: 442370_0

Title: Relaxation of mitochondrial hyperfusion in the diabetic retina via N6-furfuryladenine confers neuroprotection regardless of glycaemic status.

Authors: Aidan Anderson, Nada Alfahad, Dulani Wimalachandra, Kaouthar Bouzinab, Paula Rudzinska, Heather Wood, Isabel Fazey, Heping Xu, Timothy J. Lyons, Nicholas M. Barnes, Parth Narendran, Janet M. Lord, Saaeha Rauz, Ian G. Ganley, Tim M. Curtis, Graham R. Wallace and Jose R. Hombrebueno

General Comments:

Strengths –

1. The authors present a well-written manuscript that has clearly stated the rationale for the study, with convincing Results and an informative Discussion. The models utilized include cultured Müller cells, diabetic mice and human retinal tissues. The broad range of techniques used to measure the metabolic changes and levels of neuronal stress are appropriate. Finally, they screen four different drugs to determine which might have beneficial effects to improve the clinical parameters and molecular profiles that results in neuroprotection in their models. Their data demonstrate the role of mitochondrial quality control (MQC) in neuronal degeneration and offer a drug category that may have protective effects in human diabetic subjects.
2. Description of the Methods is detailed and thorough.

3. The references are extensive and appropriate.

4. The information will be of interest to researchers involved with mechanisms of neurodegeneration that is associated with diabetic retinopathy. The authors make an effort to correlate their tissue culture and animal findings to human diabetic retinal diseases.

Overall, it will be a valuable contribution to the field.

Weakness –

1. The article would benefit by having a list of abbreviations and description of some terms. Some readers may not be familiar with some of the terms used in the manuscript.

Specific Comments:

1. In some of the figures the fonts for the graphs are very small, making it difficult to read. It would be helpful to increase the font size.

2. Results. Line 122-133. It would be helpful to state the stage of retinal disease for the human tissues used. Also verify that the Normal tissues did not have a history of other ocular diseases.

3. Line 379. The authors use the term 'holistic approaches' but it is not clearly defined what they mean by holistic. Please clarify.

Response to Reviewers (Manuscript No. NCOMMS-23-32600-T)

Title: Relaxation of mitochondrial hyperfusion in the diabetic retina via N6-furfuryladenosine confers neuroprotection regardless of glycaemic status

We thank the Reviewers for their detailed and helpful comments. We have carefully considered the points raised and as far as possible, we have carried out additional experiments in line with the comments and revised the manuscript accordingly. The changes have been underlined in the revised manuscript and our responses are detailed below:

Referee 1

1) ABSTRACT- Line 35 and 37: The word “restored” should be replaced by “prevented”.

Response: We acknowledge the comment. However, we opted to replace the text by “*enhanced mitochondrial turnover*”, since “*prevented mitochondrial turnover*” could be mistakenly perceived as our pharmacological treatment is inhibiting the process (Abstract lines 33-34).

2) ABSTRACT- Line 38: “human-relevant” should be replaced by “relevant”. This should be corrected throughout the manuscript.

Response: We acknowledge the comment and these changes were made throughout the manuscript as suggested.

3) INTRODUCTION - The authors could mention the reference (PMID: 17704349) in which the neurodegenerative process was clearly demonstrated in the human retina, and reference (PMID: 30030554) in which a recapitulation of the main events of neurodegeneration were given.

Response: We agree these are important studies that would help the reader contextualize the neurodegenerative component of diabetic retinopathy. These have been included in the

introduction (page 3 line 55-56 and references 6-7) and also in the discussion (Page 17 line 402-404).

4) RESULTS - GFAP overexpression is a characteristic of glial activation in human diabetic retina. I presume that the authors do not measure GFAP in the Ins2-AKITA mice because GFAP overexpression is not found in this model. This is a limiting factor that should be commented in this section, as well as in the discussion.

Response: As highlighted by the reviewer and in contrast to other intermediate filaments (Vimentin; Fig 4a-b), no upregulation of GFAP toward the outer retina is consistently observed in Ins2^{Akita} mice (references 22 and 29), which is certainly a limitation of the model, as highlighted by the reviewer. This information has been added to results (Page 10 lines 229-230 and 232-233) and discussion (Page 17 lines 402-404).

5) RESULTS - Parameters of neural apoptosis should be added to the results

Response: We appreciate the constructive comment. As a parameter of neural apoptosis, we performed immunostaining against active caspase-3 (Clone C92-605, BD Biosciences) in WT and Ins2^{Akita} diabetic mouse groups. Interestingly, no substantial caspase-3 activation was found in diabetic mouse retinas for any of the groups (untreated, vehicle and KR-treated, see representative image below). This was in contrast to positive controls, where active caspase-3 (arrowheads) was clearly observed in the pyknotic nuclei of ischemic retinal neurons of oxygen-induced retinopathy (OIR) mice (postnatal stage [P]14). Other hallmarks of neuronal apoptosis (e.g., pyknotic nuclei) were also not particularly evident by 8-months of diabetes in Ins2^{Akita} mice, as found in positive OIR controls (arrowheads).

Supporting the lack of substantial apoptosis in Ins2^{Akita} mice and as previously reported by our group (PMID: 24848689), neuronal deterioration in this model is mainly associated with deterioration of neuronal structures (e.g., cone-photoreceptor segments and synapses; Fig. 6m-o), as it also underpins human diabetes (Fig 1c-h). These data also suggest that neuronal loss may progress slowly during diabetes and, therefore, cellular apoptosis may not be substantially elevated at a given time-point. Nevertheless, our study provided clear evidence for the potential

of KR to prevent neuronal deterioration as assessed by multiple means, including in vivo (SD-OCT – Fig 6j), at the functional level (ERG - Fig 6k-l), and post-mortem (Fig 6m-p).

Retinal photomicrographs of active caspase-3 in 8-month diabetic Ins2^{Akita} and Oxygen induced retinopathy (Postnatal stage [P]14) mice. Arrowheads indicate active caspase-3 in pyknotic nuclei. ONL, outer nuclear layer; OPL, outer plexiform layer; INL, inner nuclear layer; IPL, inner plexiform layer; GCL, ganglion cell layer.

6) RESULTS - A direct measurement of KR in the retina Ins2-AKITA mice would be of relevance to strength the results.

Response: We appreciate the constructive comment, however we believe that the existing data provide good evidence of drug-target engagement in the relevant tissue (i.e., retina) at a pharmacologically relevant concentration. In this regard, KR (90 mg/Kg/day) was demonstrated to be effective in increasing mitophagy and TFAM+ mitochondrial nucleoids in Mito-QC Ins2^{Akita} retinas (Fig 6b-c, g). This effect was corroborated by long-term oral administration of KR in Ins2^{Akita} mice (Supplemental Fig 9). Our in vitro data further supports

such drug-target interaction, as KR elicited an equivalent biological action (i.e., increased mitophagy and TFAM) in the context of mitochondrial hyperfusion (Fig 5c and Supplemental Fig 8). As explained in the Discussion (page 18 lines 417-428), our future goal will be to accurately understand the pharmacokinetics and ADME (absorption, distribution, metabolism, and excretion) of KR. These experiments will require extensive resources (HLPC, GC-MS, or Bioluminescence/NIR imaging [IVIS]) given the technical challenges involved in detecting KR and its different metabolites, as well as optimizing the use of new animals for that specific purpose.

7) DISCUSSION - The limitations of the study, in particularly the reproducibility of the results in other experimental models and the lack of evaluation of potential systemic toxicity of systemic administration of KR should be commented on in this section.

Response: We appreciate the constructive comment. Limitations have now been added to the Discussion as suggested (Page 18 lines 417-418 and 422-423).

Referee 2

1) The article would benefit by having a list of abbreviations and description of some terms. Some readers may not be familiar with some of the terms used in the manuscript.

Response: We appreciate the constructive comment and described abbreviations that were not included initially (PGC-1 α , PINK1, M-opsin, GFAP, ILs, MCP-1, VEGF-A, GM-CSF, DMSO and DMEM). In accordance with journal guidelines, we defined abbreviations at first mention.

2) In some of the figures the fonts for the graphs are very small, making it difficult to read. It would be helpful to increase the font size.

Response: As suggested, we have edited all main and supplementary figures to adjust font size. We also edited the presentation of figures to reflect single data points (where necessary) in accordance with editorial guidelines.

3) Results. Line 122-133. It would be helpful to state the stage of retinal disease for the human tissues used. Also verify that the Normal tissues did not have a history of other ocular diseases.

Response: We appreciate the constructive comment and this information has been updated in the Results (page 6, lines 125-126) and Methods (page 21, lines 499-503) sections to clarify the use of male and female diabetic donors (n=6 with without clinical retinopathy; 2 with mild NPDR) and non-diabetic donors (n = 3, without a history of ocular disease, as further supported at the histological level by our neuronal study - Fig 1). We also clarified that all diabetic donors included in the study did not receive any treatment for DR (e.g., laser photocoagulation and/or intraocular injections).

4) Line 379. The authors use the term 'holistic approaches' but it is not clearly defined what they mean by holistic. Please clarify.

Response: We appreciate the constructive comment. This sentence has been edited for better comprehension now reading “*Instead, approaches that restore both mitochondrial dynamics (fusion/fission) and turnover (mitophagy and biogenesis) in a controlled fashion*” (see Page 16 lines 382-384).

REVIEWERS' COMMENTS

Reviewer #1 (Remarks to the Author):

The authors have answered correctly my comments.